# Soccer goalkeeper expertise identification based on eye movements

**Benedikt W. Hosp**[1,2]*, **Florian Schultz**[2], **Oliver Höner**[2], **Enkelejda Kasneci**[1]

**1** Human-Computer Interaction, University of Tübingen, Tübingen, Germany, **2** Institute of Sport Science, University of Tübingen, Tübingen, Germany

* benedikt.hosp@uni-tuebingen.de

## Abstract

By focusing on high experimental control and realistic presentation, the latest research in expertise assessment of soccer players demonstrates the importance of perceptual skills, especially in decision making. Our work captured omnidirectional in-field scenes displayed through virtual reality glasses to 12 expert players (picked by DFB), 10 regional league intermediate players, and13 novice soccer goalkeepers in order to assess the perceptual skills of athletes in an optimized manner. All scenes were shown from the perspective of the same natural goalkeeper and ended after the return pass to that goalkeeper. Based on the gaze behavior of each player, we classified their expertise with common machine learning techniques. Our results show that eye movements contain highly informative features and thus enable a classification of goalkeepers between three stages of expertise, namely elite youth player, regional league player, and novice, at a high accuracy of 78.2%. This research underscores the importance of eye tracking and machine learning in perceptual expertise research and paves the way for perceptual-cognitive diagnosis as well as future training systems.

**Data Availability Statement:** Eye tracking data are available from: https://atreus.informatik.uni-tuebingen.de/hosp/goalkeeperexpertisesupvervisedml_dataset.

## 1 Introduction

Along with physical performance, perceptual-cognitive skills play an increasingly important role as cognitive performance factors in sport games [1–4]. In perceptual research examining the underlying processes of these skills, subjects are typically placed in a situation where they have to react while their behavior is being recorded and subsequently analyzed. Such behavior can be assigned to a class to provide information about performance levels [5, 6]. Many studies in sports, and soccer in particular [1–4, 7–9], have shown that athletes in a high performance class have a higher level of perceptual-cognitive skill leading to greater success in their respective sports. However, this research still faces challenges, especially related to experimental control and a representative presentation of the situation being studied [10]. Furthermore, the potential of novel technologies such as eye tracking to assess the underlying perceptual-cognitive processes has not yet been fully exploited, particularly in regard to the analysis of complex eye-tracking data [11]. In this work, we research how to handle and analyze large and complex eye-tracking data in an optimized manner. We achieve this by applying common supervised

**Funding:** The work was partly funded by the German Football Association (DFB). There was no additional external funding received for this study.

**Competing interests:** The authors have declared that no competing interests exist.

machine learning techniques to the gaze behavior of soccer goalkeepers during a decision-making task in build-up game situations that are presented as 360˚-videos in a consumer-grade virtual reality headset.

The latest sports-scientific expertise research shows that, when it comes to decision-making, experts have more efficient gaze behavior because they apply an advanced cue utilization to identify and interpret relevant cues [12]. This behavior enables experts to make more efficient decisions than non-experts, e.g. in a situation such as a goalkeeper facing game build-up. From both a scientific and practical sports perspective, factors that lead to successful perception and form expertise are of particular importance. To measure perception based expertise, a diagnostic system that recognizes expertise and provides well-founded information about the individual attributes of perception is needed. These attributes are usually considered in isolation allowing for the specific recognition of their influence on expertise. To permit athletes to apply their natural gaze behavior, the experimental environment is important. However, one of the main problems in perceptual cognitive research persists: realism vs. control. In a meta review of more than 60 studies on natural gaze behavior from the last 40 years, Kredel et al. [10] postulate that one of the main challenges in perception research lies in a trade-off between experimental control and a valid, realistic presentation. Diagnostic and training models are often implemented or supported by digital means.

This is nothing new. In sports psychological research, new inventions in computer science such as presentation devices (i.e. CAVE [13], virtual reality head-mounted displays (VR-HMD) [14]), interface devices (i.e. virtual reality trackers, leap motion etc.), or biometric feature recognition devices (i.e. eye tracker [15]) are used more and more often. As a new upcoming technology, virtual reality (VR) devices are used more frequently as stimulus presentation and interaction devices. Accordingly, a fundamental aspect of perception research is a highly realistic presentation mode, which allows for natural gaze behavior during diagnostic. VR technology makes this possible by displaying realistic, immersive environments. However, this strength, facilitating natural gaze behaviour, comes less from the VR technology itself. According to Gray [16], the degree to which the perceptual-cognitive requirements of the real task are replicated in such environments depends on psychological fidelity. Next to immersion and presence, Harris et al. [17] propose the expansion of a simulation characterization into a typology of fidelity, also containing psychological fidelity, to determine the realism of a simulation. VR offers an immersive experience through the use of 4k 360˚ video in HMDs. This technology offers a higher level of realism than other systems, such as CAVE systems, by providing higher levels of psychological fidelity [16, 17]. VR is therefore a popular and optimal tool for perception research. Bideau et al. [18] summarize further advantages of VR in their work. Their main contribution, however, is their immersive virtual reality that elicits expert responses similar to those in the real world.

In a narrower sense, VR is based on computer generated imagery (CGI). One advantage of fully CGI-based environments is the potential of user interaction with the environment, which presumably increases the immersive experience. On the other hand, fully CGI-based environments contain moving avatars that are not always natural in appearance and often hide environmental influences. This might prevent high immersion and could influence the participant's gaze behaviour. Therefore, we chose a realistic environment with 360˚ stimuli in our work to provide a nearly natural environment that does not influence the participant's gaze behaviour. As this work focuses on the cognitive processes of decision-making, we concentrate less on realistic interaction methods.

Especially noteworthy are the developments of VR devices with regard to integrated measuring devices. More and more devices integrate eye trackers directly. This, in combination with a photo-realistic environment in VR glasses, allows for the measurement of almost

optimal user gaze behavior while also exhibiting highly realistic stimuli. Eye trackers provide a sound foundation with high temporal and spatial resolution for the research of perceptual processes. The combination of VR and high speed eye tracking allow for the collection of a massive amount of highly complex data. The high quality eye images and freedom of movement inherent in a mobile eye tracker in combination with the high speed of a remote eye tracker, control over the stimulus in lab setting (VR), and realism of in-situ stimuli by omnidirectional videos, leads to a highly complex outcome. Analysis of such data is a particular challenge, indicating the need for new analysis methods. As we want to infer underlying mechanisms of perceptual-cognitive expertise, tracking eye movements is our method of choice in this work. Generally, perceptual research focuses on eye tracking because, as a direct measuring method, it allows for a high degree of experimental control [19–22]. Besides a realistic presentation and high degree of experimental control, VR can also be used to model the perception [23] of athletes and thus creates a diagnostic system. A diagnostic system has the ability to infer the current performance status of athletes to identify performance-limiting deficits, an interesting provision of insight for the athletes and their coaches as well. Most importantly, such a diagnostic system forms the basis for an adaptive, personalized, and perceptual-cognitive training system to work on the reduction of these deficits.

Thus far, eye tracking studies have focused on either in-situ setups with a realistic presentation mode and mobile eye trackers (field camera showing the user's field of view) or on laboratory setups with high experimental control using remote eye trackers [24–29]. Since mobile eye trackers are rarely faster than 100-120 Hz, saccades and smooth pursuits cannot be properly detected at such a speed. Investigations in an in-situ context are, therefore, limited to the observation of fixations. Fixations are eye movement events during which the eye is focused on an object for a certain period of time, and thus projects the object onto the fovea of the eye, so that information about the object can be cognitively processed. The calculation of fixations with such a slow input signal leads to inaccuracies in the recognition of the start and end of a fixation. Only limited knowledge can be gained using such eye trackers because additional information contained in other eye events, such as saccades and smooth pursuits, cannot be correctly computed. This prevents the use of these eye trackers as robust expert measurement devices. Saccades are the jumps between fixations that allow the eye to realign. They can be as fast as 500˚/s. Smooth pursuits are particularly significant in ball sports because they are fixations on moving objects i.e. moving players. However, especially in perception studies in soccer in VR-like environments, slow eye trackers with about 25-50 Hz are primarily used [30–33]. This speed limits the significance of these studies to fixation and attention distribution in areas of interest (AOI). Aksum et al. [32], for example, used the Tobii Pro Glasses 2 with a field camera set to 25 Hz. Therefore, only fixations or low speed information were available and there was no equal stimuli for comparable results between participants. In a review of 38 studies, McGuckian et al. [34] summarized the eye movement feature types used to quantify the visual perception and exploration behaviour of soccer players. Except for Bishop et al. [35], all studies were restricted to fixations thus restricting the knowledge that could be gained by studying other eye movement features. The integration of high speed eye trackers into VR glasses combines both strengths: high experimental control of a high speed eye tracker and a photo-realistic stereoscopic VR environment.

As eye trackers are used more frequently and more accurate, faster, and ubiquitous devices become available, huge amounts of precise data from fixations, saccades, and smooth pursuits can be generated. This, however, cannot be handled in entirety utilizing previous analysis strategies. Machine learning has the power to deal with large amounts of data. In fact, machine learning algorithms typically improve with more data and allow, by publishing the model's parameter set, fast, precise, and objective reproducible ways to conduct data analysis. Machine

learning methods have already been successfully applied in several eye tracking studies. Expertise classification problems in particular can be solved using such methods as shown by Castner et al. in dentistry education [19, 36] and Eivazi et al. in microsurgery [20, 37–39]. Machine learning techniques are the current state-of-the-art for expertise identification and classification. Both supervised learning algorithms [36, 37] and unsupervised methods or deep neural networks [19] have shown their proclivity for this kind of problem solving. This combination of eye tracking and machine learning is especially well suited when it comes to subconscious behaviors such as eye movements features. These methods have the potential to greatly benefit the discovery of different latent features in gaze behavior and their relation and significance to expertise classification.

In this work, we present a model for the recognition of soccer goalkeepers' expertise when making decisions in build-up game situations by means of machine learning algorithms that rely solely on eye movements. We also present an investigation of the influence of single features on explainable differences between single classes. This pilot study is meant to be a first step towards a perceptual-cognitive diagnostic system and a perceptual-cognitive virtual reality training system, respectively.

## 2 Method

### 2.1 Experimental setup

In this study, we employed an HTC Vive, a consumer-grade virtual reality (VR) headset. Gaze was recorded through integration of the SMI high speed eye tracker at 250 Hz. The SteamVR framework is an open-source software that interfaces common real-time game engines with the VR glasses to display custom virtual environments. We projected omnidirectional 4k footage on the inside of a sphere that envelopes the user's field of view, leading to high immersion in a realistic scene.

**2.1.1 Stimulus material.** We captured the 360°-footage by placing an Insta Pro 360 (360°-camera) on the soccer field keyed to the position of the goalkeeper. Members of a German First League's elite youth academy were playing 26 different 6 (5 field players + goalkeeper) versus 5 match scenes on one half of a soccer field. Each scene was developed with a training staff team from the German Football Association (DFB) and each decision was ranked by this team. There were 5 options (teammates) plus one "emergency" option (kick out). For choosing the option rated the best by the staff team because it ensured continuation of the game, the participant earned 1 point. All other options were rated with 0 points. Conceptually, all videos had the following content: The video starts with a pass by the goalkeeper to one of the teammates. The team passes the ball a few times until the goalkeeper (camera position) receives the last return pass. The video stops after this last pass and a black screen is presented. The participant now has 1.5 seconds time to report which option they've decided on and the color of the ball that was printed on the last return pass (to force all participants to recognize the last return pass realistically).

**2.1.2 Participants.** We collected data from 12 German expert youth soccer goalkeepers (U-15 to U-21) during two youth elite goalkeeper camps. The data from 10 intermediates was captured in our laboratory and comes from regional league soccer goalkeepers (semi-professional). Data from 13 novices came from players with up to 2 years of experience, no participation in competitions, and no training on a weekly basis. The experts train 8.83 hours per week and are an average age of 16.6 years old. They have actively played soccer for about 9 years which is significantly more than the novices (1.78 years), but less than the intermediates (15.5 years). This may be a result of their age difference. The intermediates are 22 years old on

**Table 1. Participants summary.**

| Class | Attribute | Participants | |
|---|---|---|---|
| | | *Average* | *Std. Dev.* |
| Experts | Age | 16.60 | 1.54 |
| | Active years | 9.16 | 5.04 |
| | Training hours/week | 8.83 | 4.27 |
| Intermediates | Age | 22.00 | 3.72 |
| | Active years | 15.50 | 5.77 |
| | Training hours/week | 4.94 | 0.91 |
| Novices | Age | 28.64 | 3.72 |
| | Active years | 1.78 | 5.21 |
| | Training hours/week | 0.00 | 0.00 |

average, but train nearly half the number of hours per week as compared to the experts. Characteristics of the participants can be seen in Table 1.

**2.1.3 Procedure.** The study was confirmed by the Faculty of Economics and Social Sciences Ethic Committee of the University of Tübingen. After signing a consent form to allow the usage of their data, we familiarized the participants with the footage.

The study contained two blocks consisting of the same 26 stimuli in each (conceptually as mentioned in the stimulus material section). The stimuli in the second block were presented in a different randomized order. Each decision made on the continuation of a video has a binary rating as only the best decision counted as 1 (correct) while all other options were rated 0 (incorrect). At first, 5 different sample screenshots (example view see Fig 1 in equirectangular form or S1 Video 4.1 for a cross section of the stimulus presentation sphere) and the corresponding sample stimuli were shown and explained to acclimate participants with the setup. To learn the decision options, we also showed a schematic overview before every sample screenshot (see Fig 2).

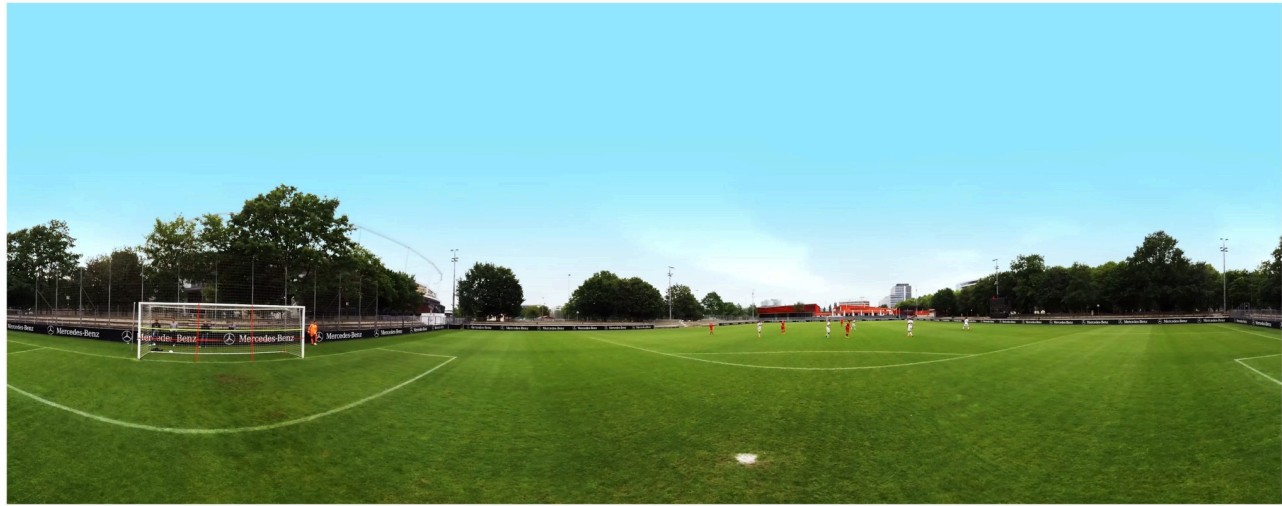

**Fig 1. Example stimulus in equirectangular format.**

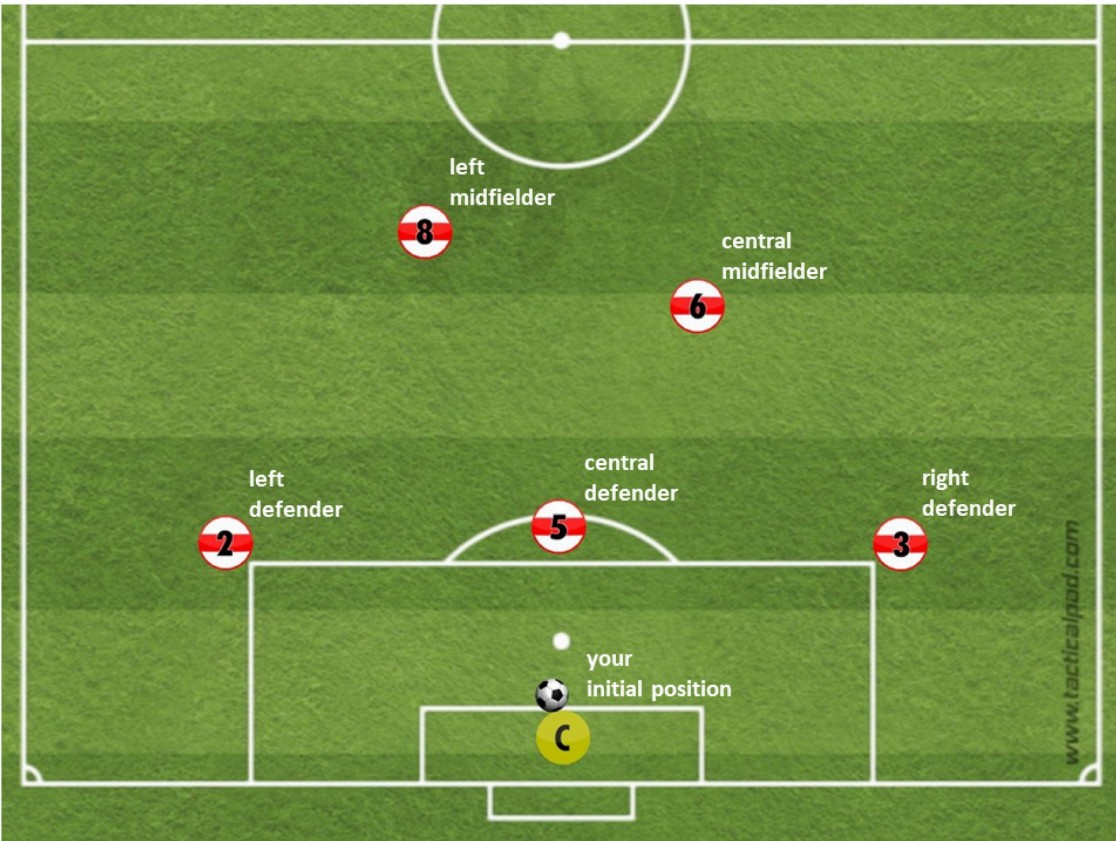

**Fig 2. Schematic overview of the response options.** Emergency option kick out is not shown.

## 2.2 Eye tracking

The raw data of the SMI Eye tracker can be exported from the proprietary BeGaze software as csv files. BeGaze already provides the calculation of different eye movement features based on raw gaze points. As we receive high speed data from the eye tracker, we use built-in high speed event detection. The software first calculates the saccades based on the peak threshold, which means the minimum saccade duration (in ms) varies and is set dependent on the peak threshold default value of $40°/s$. In a second step, the software calculates the fixations. Samples are considered belonging to a fixation when they are between a saccade or blink. With a minimum fixation duration of 50 ms, we reject all fixations below this threshold. As there is no generally applicable method for the detection of smooth pursuits, this type of event is included and encoded as a fixation with longer duration and wider dispersion. We marked fixations with a fixation dispersion of more than 100 px as smooth pursuits. By doing this, we split fixations into normal length fixations and long fixations, the latter considered and referred to as smooth pursuits. Since there is no information about the pixel size of the HMD in mm, it is hard to define a robust threshold in pixels. Therefore, this threshold is an empirical value based on the typical length of the player's routes as main stimuli in the video. The following section describes the steps that are necessary to train a model based on these eye movement features.

**2.2.1 Feature selection.** Since it is not completely clear which subset of eye movement features explain differences in expertise, we pursued a brute-force method. All possible measures

issued by the eye tracking device were considered and their importance subsequently evaluated. For the classification of expertise level we focused on the following features:

- event duration and frequency (fixation/saccade),

- fixation dispersion (in °),

- smooth pursuit duration (in ms)

- smooth pursuit dispersion (in °)

- saccade amplitude (in °),

- average saccade acceleration (in $°/s^2$),

- peak saccade acceleration (in $°/s^2$),

- average saccade deceleration (in $°/s^2$),

- peak saccade deceleration (in $°/s^2$),

- average saccade velocity (in $°/s$),

- peak saccade velocity (in $°/s$).

Each participant viewed all 26 stimuli two times, resulting in 52 trials per subject. First, we viewed the samples from these 52 trials and checked the confidence measures of the eye tracking device. We removed all trials with less than 75% tracking ratio, as gaze data below this threshold is not reliable. Due to errors in the eye tracking device, not all participant data is available for every trial. Table 2 shows an overview of the lost trials. For two participants, 11 trials had a low tracking ratio. On participant 18, we lost 35 trials. On participant 33, one trial was lost. This results in 1658 out of 1716 valid trials in total. 3.3% of the trials were lost due to eye tracking device errors.

**2.2.2 Data cleaning.** We checked the remaining data for the quality of saccades. This data preparation is necessary to remove erroneous and low quality data that comes from poor detection on behalf of the eye tracking device and does not reflect the correct gaze. We investigated invalid samples and removed (1) all saccades with invalid starting position values, (2) all saccades with invalid intra-saccade samples, and (3) all saccades with invalid velocity, acceleration, or deceleration values.

(1) Invalid starting position: 0.22% saccades started at coordinates (0;0). This is an encoding for an error of the eye tracking device. As amplitude, acceleration, deceleration and velocity are calculated based on the distance from start- to endpoint, these calculations result in physiological impossible values, e.g., over 360° saccade amplitudes.

(2) Invalid intra-saccade values: Another error of the eye tracking device stems from the way the saccade amplitude is calculated through the average velocity (Eq 1) which is based on

**Table 2. Overview of the amount of erroneous trials, based on eye tracking device errors.**

| Overview erroneous trials | |
|---|---|
| **Participant** | **Number of invalid trials** |
| 1 | 11 |
| 8 | 11 |
| 18 | 25 |
| 33 | 1 |
| all others | 0 |

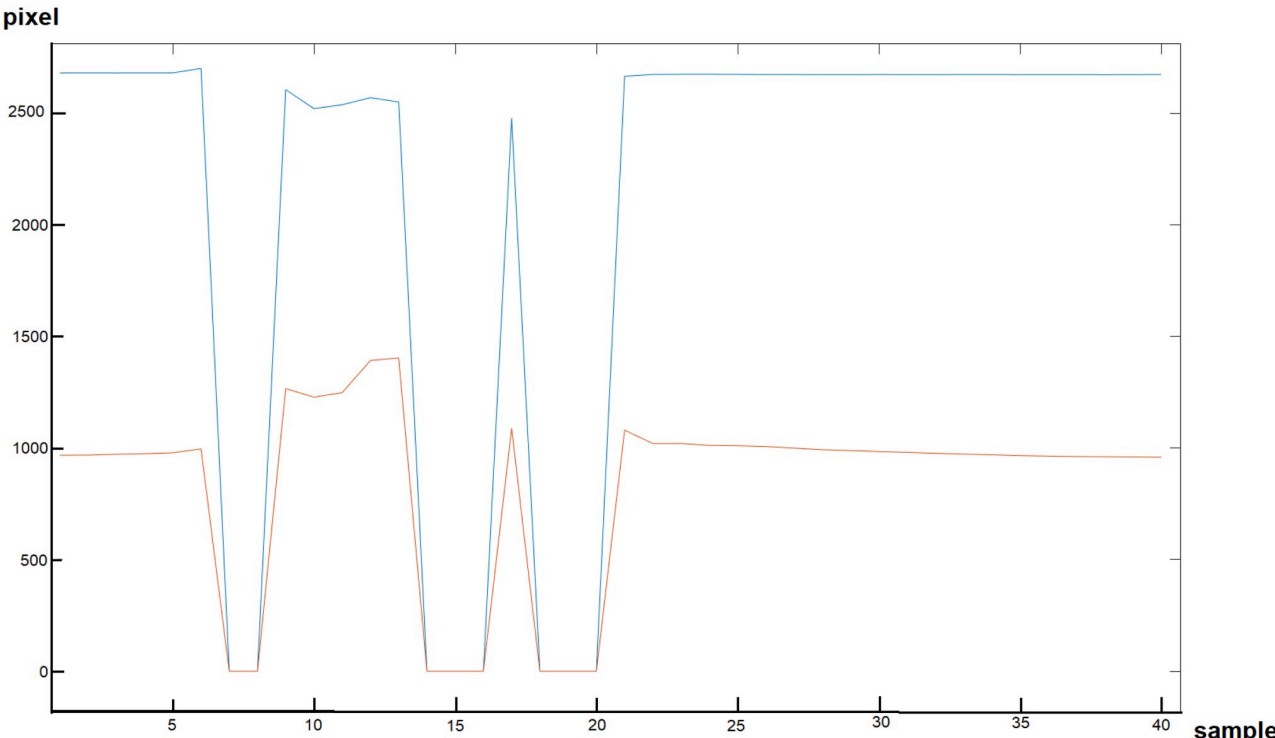

**Fig 3. Example of invalid intra-saccade values.** The x-axis shows the number of the gaze signal sample (40 samples, 250 Hz, 160 ms duration) and the y-axis shows the position in pixel. The blue line represents the x-signal of the gaze and the orange line the y-signal.

the distance of the mean of start and endpoints on a sample-to-sample basis (see Eq 2). 3.6% of the saccades had at least one invalid gaze sample and were removed (example see Fig 3).

$$\oslash Velocity * EventDuration \tag{1}$$

$$\frac{1}{n} * \sum_{1}^{n} \frac{dist(startpoint(i), endpoint(i))}{EventDuration(i)} \tag{2}$$

On Fig 3, the gaze signal samples 7, 8, 14-16, 18-20 (x-axis) both, the x- and y-signal (blue and red line, respectively) show zero values and thereby indicate a tracking loss. As the saccade amplitude is based on the average velocity which is calculated on a sample-to-sample Eq 2, the velocity from samples 6 to 7, 8 to 9, 13 to 14, 16 to 17, 17 to 18, and 20 to 21 significantly increase the average velocity as the distances are high (on average over 2400 px for x-signal and over 1000px for y-signal, which corresponds to a turn of 225° on x-axis and 187.5° on y-axis in the time of 4 ms between two consecutive samples).

There are two interpretations for saccadic amplitude. The first refers to the shortest distance from start to end point of a saccadic movement (i.e., a straight line) and the second describes the total distance traveled along the (potentially curved [40], p.311) trajectory of the saccade. The SMI implementation follows the second definition. We could have potentially interpolated invalid intra-saccade samples instead of completely removing the complete saccade from analysis, but this leads to uncertainties that can affect the amplitude depending on the number of invalid samples and does not necessarily represent the true curvature of the saccade.

(3) As the velocity increases as a function of the saccade amplitude [41], 4.8% of the saccades were ignored because of the restriction on velocities greater than 1000°/s. Similar to extreme velocities, we removed all saccade samples that exceeded the maximum theoretical acceleration and deceleration thresholds. Saccades with longer amplitudes have higher velocity, acceleration, and deceleration, but can not exceed the physiological boundaries of $100.000°/s^2$ [40]. 4.0% of all saccades exceeded this limit. As most of the invalid samples had more than one error source, we only removed 5.5% of the saccades (3.5% of all samples) in total.

After cleaning the data, we used the remaining samples to calculate the average, maximum, minimum, and standard deviation of the features. This resulted in 36 individual features. We use those for classifying expertise in the following.

## 2.3 Machine learning model

In the following, we refer to *expert samples* as trials completed by an elite youth player from a DFB goalkeeper camp, *intermediate samples* as those completed by regional league players, and *novice samples* as those completed by amateur players. We built a support vector machine model (SVM) and validated our model in two steps: cross-validation and leave-out validation. We trained and evaluated our model in 1000 runs with both validations. For each run, we trained a model, validated with cross-validation, with samples from 8 experts, 8 intermediates, and 8 novices. We used samples from two participants representing each group of those remaining to predict their classes (leave-out validation). The expert as well as the intermediate and novice samples in the validation set were picked randomly for each run.

**2.3.1 Sample assignment.** We found that the way in which the data set samples are divided into training and evaluation sets is key and a participant-wise manner should be applied. By randomly picking samples independent of the corresponding participant, participant samples are usually distributed on both the training and the evaluation sets (illustrated in Fig 4). This leads to an unexpected learning behavior that does not necessarily classify expertise directly. Rather, the method matches the origin of a sample to a specific participant thereby indirectly identifying that participant's level of expertise. This means that a model should work perfectly for known participants, but is unlikely to work for unseen data. Multiple studies show that human gaze behavior follows idiosyncratic patterns. Holmqvist et al. [40] show that a significant number of eye tracking measures underlie the participants' idiosyncrasy which also means that inter-participant differences are much higher than intra-participant

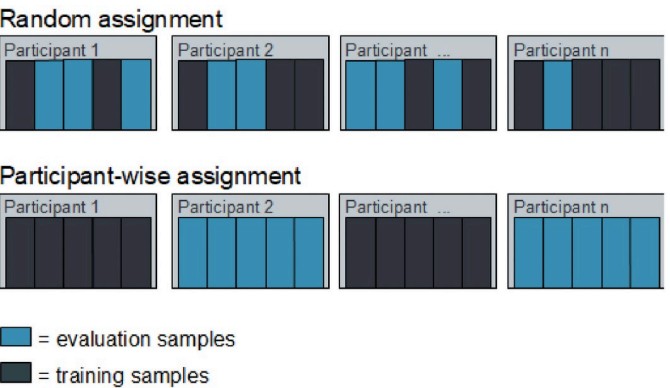

**Fig 4. Example sample assignment.** Top row shows a random assignment of samples, independent of the corresponding participant. Bottom row shows participant-wise sample assignment to training and evaluation set.

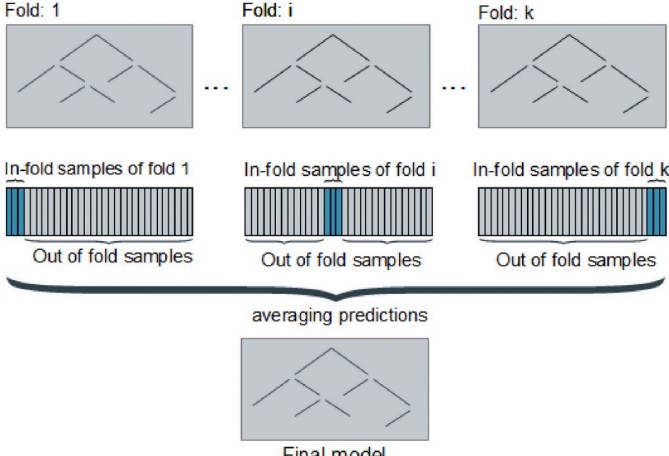

**Fig 5. Illustration of the k cross-validation procedure.** Each of the k models has a different out-of-fold and in-fold data set. We build the final model on the average of all predictions from all k models.

differences. A classifier learns a biometric, a person-specific measure, instead of an expertise representation.

**2.3.2 Model building.** To find a model that is robust to high data variations, we applied a cross-validation during training. The final model is based on the average of k = 50 models, with k = number of folds in the cross-validation. For each model $m_i$, with $i \in \{1, \ldots, k\}$, we used all out-of fold data of the i-th fold to train and evaluate $m_i$ with the in-fold data of the i-th fold (this procedure is illustrated in Fig 5). The final model was evaluated with a leave-out validation. The cross-validation step during training is independent from the leave-one-out validation which requires entirely new data that the model has never seen before. Information from cross-validation is used during the building and optimizing of the model while leave-one-out validation only provided information about the model's prediction accuracy when using completely new data.

With a total of 810 valid samples, equally distributed between expert, intermediate and novice samples, we built a subset of 552 samples for training the model and a subset of 258 samples for evaluation. As each sample represents one trial, our approach here was to predict whether a trial belongs to an expert, intermediate or novice class. We tested assumption in different approaches.

**2.3.3 Classifiability.** First, we used all 46 features to check the classifiability of this kind of data. The first approach contains all features from section *Feature selection* 2.2.1 with their derivations, (namely average, maximum, minimum, and standard deviation) to build an SVM model (Tables 3–5 show all features with their derivations, divided by class). When the binary case (expert vs. intermediates) results point out the ability of classification, the ternary case (expert vs. intermediate vs. novice) should be investigated.

**2.3.4 Significant features.** Second, we looked at the features themselves, checking for differences between the single features according to their class as well as the significance level of feature differences under 0.11%. We built a model based on the features with a significance level under 0.11% (Tables 3–5 all white cells, gray cells indicate that there is no significant difference between the groups).

**2.3.5 Most frequent features.** In a third approach, we reduced the number of features by running the prediction on all 46 features 1000 times. By taking the most frequent features in the model, we searched for a subset of features that would prevent the model from overfitting

**Table 3. All 46 features with their derivations.** Novice class.

| Novices | | | | |
|---|---|---|---|---|
| Features | average | std. dev. | minimum | maximum |
| **Fixation** | | | | |
| frequency (Hz) | 0.214 | - | - | - |
| duration (ms) | 214.017 | 31.926 | 190.49 | 239.30 |
| dispersion (pixels) | 72.092 | 25.68 | 24.67 | 110.523 |
| **Saccade** | | | | |
| frequency (Hz) | 0.071 | - | - | - |
| duration (ms) | 71.688 | 38.869 | 26.514 | 175.460 |
| amplitude (°) | 9.294 | 9.417 | 0.574 | 51.402 |
| **Saccade mean acceleration** | | | | |
| mean ($°/s^2$) | 4263.381 | 2482.019 | 366.666 | 13984.563 |
| peak ($°/s^2$) | 9322.483168 | 5777.273817 | 231.836 | 28355.224 |
| **Saccade deceleration** | | | | |
| peak ($°/s^2$) | -6848.104 | 4166.262 | -35563.646 | -411.760 |
| **Saccade velocity** | | | | |
| mean ($°/s$) | 105.463 | 65.023 | 20.288 | 298.134 |
| peak ($°/s$) | 215.245 | 129.294 | 40.310 | 766.157 |
| **Smooth pursuit** | | | | |
| duration (ms) | 302.637 | 278.112 | 75.629 | 1026.329 |
| dispersion (pixels) | 622.805 | 201.268 | 185.437 | 1085.903 |

Gray cells show features with no significant differences between classes. Orange cells stand for a most frequent feature.

**Table 4. All 46 features with their derivations.** Intermediate class.

| Intermediates | | | | |
|---|---|---|---|---|
| Features | average | std. dev. | minimum | maximum |
| **Fixation** | | | | |
| frequency (Hz) | 0.255 | - | - | - |
| duration (ms) | 255.225 | 53.379 | 215.835 | 299.623 |
| dispersion (pixels) | 73.173 | 26.548 | 23.070 | 114.762 |
| **Saccade** | | | | |
| frequency (Hz) | 0.084 | - | - | - |
| duration (ms) | 84.349 | 59.726 | 26.127 | 246.121 |
| amplitude (°) | 9.883 | 10.674 | 0.572 | 54.835 |
| **Saccade mean acceleration** | | | | |
| mean ($°/s^2$) | 4123.970 | 2685.991 | 315.346 | 15472.889 |
| peak ($°/s^2$) | 8920.177 | 5989.251 | 216.722 | 28266.000 |
| **Saccade deceleration** | | | | |
| peak ($°/s^2$) | -6948.491 | 4770.063 | -36334.137 | -231.355 |
| **Saccade velocity** | | | | |
| mean ($°/s$) | 104.199 | 66.682 | 21.520 | 331.111 |
| peak ($°/s$) | 213.835 | 136.529 | 40.109 | 764.027 |
| **Smooth pursuit** | | | | |
| duration (ms) | 291.092 | 278.718 | 73.835 | 977.120 |
| dispersion (pixels) | 425.089 | 124.853 | 168.320 | 694.370 |

We consider samples as belonging to a smooth pursuit when the dispersion of the samples is greater than 100 px. As the size of the players in the stimulus varies around 90 pixel + a buffer.

**Table 5. All 46 features with their derivations.** Expert class.

| Features | Experts | | | |
| --- | --- | --- | --- | --- |
| | average | std. dev. | minimum | maximum |
| **Fixation** | | | | |
| frequency (Hz) | 0.241 | - | - | - |
| duration (ms) | 241.509 | 58.629 | 198.132 | 291.721 |
| dispersion (pixels) | 72.837 | 25.989 | 21.736 | 114.549 |
| **Saccade** | | | | |
| frequency (Hz) | 0.007 | - | - | - |
| duration (ms) | 65.472 | 35.548 | 25.019 | 163.415 |
| amplitude (°) | 8.938 | 9.430 | 0.567 | 52.029 |
| **Saccade mean acceleration** | | | | |
| mean (°/$s^2$) | 4769.655 | 3064.343 | 390.094 | 18965.944 |
| peak (°/$s^2$) | 10026.456 | 7094.930 | 175.242 | 39445.125 |
| **Saccade deceleration** | | | | |
| peak (°/$s^2$) | -7912.190 | 5492.287 | -43479.916 | -362.396 |
| **Saccade velocity** | | | | |
| mean (°/$s$) | 110.675 | 72.737 | 21.182 | 375.363 |
| peak (°/$s$) | 238.371 | 157.740 | 40.262 | 935.514 |
| **Smooth pursuit** | | | | |
| duration (ms) | 276.785 | 265.679 | 74.404 | 953.660 |
| dispersion (pixels) | 399.939 | 112.414 | 336.016 | 505.031 |

and allow for illustrative results representing differences between expertise classes with a minimum number of features. These most frequent features are imperative for the model to distinguish the classes. During training, the model indicated which features were the most important for prediction in each run. The resulting features with the highest frequency (and therefore highest importance for the model) in our test can be seen in Tables 3–5, in orange.

## 3 Results

We first report the results of a classification test to determine whether gaze variations between experts are smaller than the differences between classes. Then, as we first need to identify any potential differences between experts and novices, the classifiablity test (binary classification) provides a deeper analysis on the model trained with all features for distinguishing experts and novices. The remaining chapter describes two ternary models based on a subset of features obtained through 1) their significance level and 2) their frequency in the all feature model.

### 3.1 Expert variation classification

To strengthen this work's implicit assumption that it is possible to distinguish between novices, intermediates, and experts based on gaze behavior, we evaluated expert data separately by reversing a subset of experts with intermediates. After 100 iterations in which half of the experts where randomly labeled as intermediates, the average classification accuracy was below chance-level. This means that the model can't differentiate between experts and reversed experts properly and thus strengthens our assumption that gaze variations between experts are smaller than the differences between experts, intermediates, and novices.

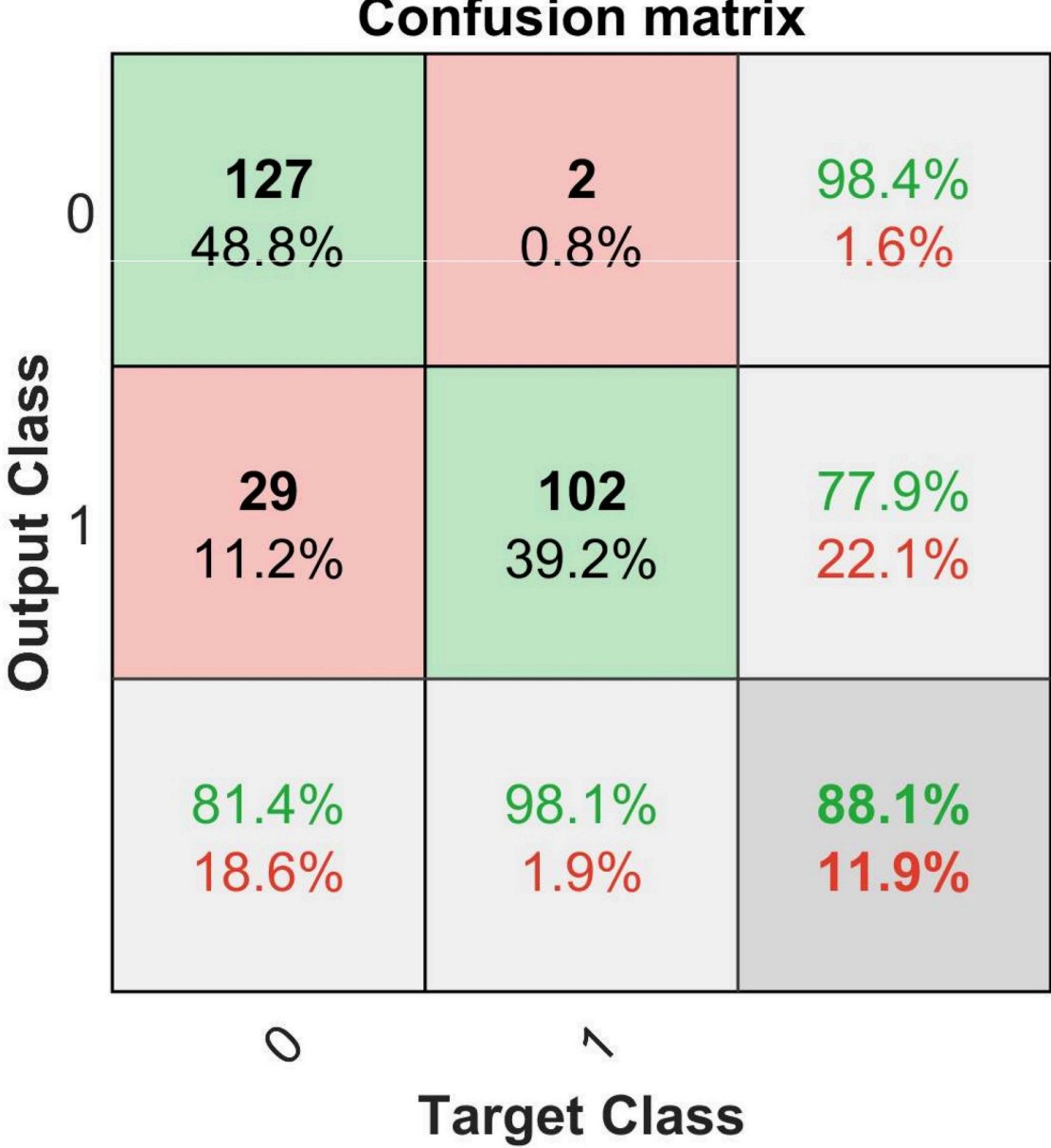

**Fig 6. Binary confusion matrix about predictions on 100 randomized runs.**

### 3.2 Binary classification

The classifiability test shows promising results. This binary model is able to distinguish between experts and intermediates with an accuracy of 88.1%. The model has a false negative rate of 1.6% and a false positive rate of 18.6%. This means the binary model predicted two out of 260 samples falsely as class zero and 29 samples that are class zero as class one. As the false negative rate is pretty low, the resulting miss rate is low (11.9%) as well. The confusion matrix (Fig 6) shows the overall metrics. The binary model is better at predicting class zero samples (intermediates) than class one samples (experts). The overall accuracy of 88.1% is sufficient to investigate on a ternary classification. In the following, we offer deeper insights into ternary

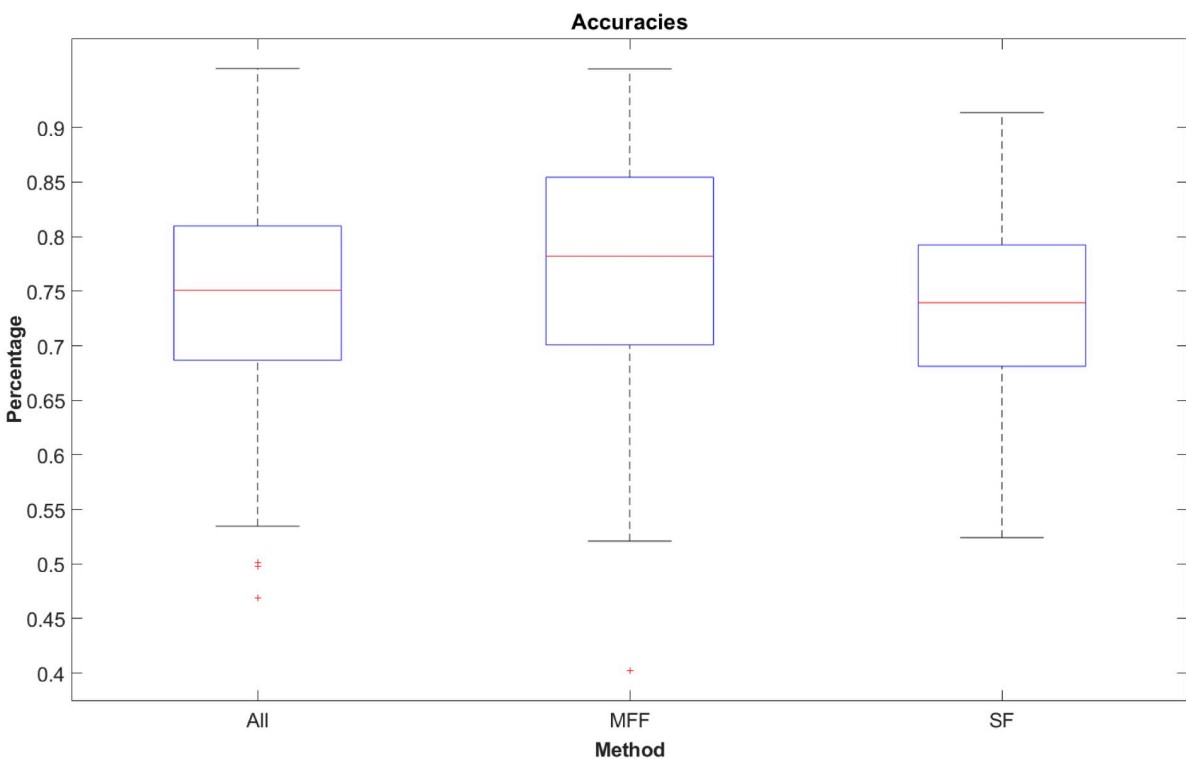

**Fig 7. Boxplot showing the accuracy values of the ternary methods.** All three models have median accuracy values $\sim 75 - 80\%$.

approaches by looking at the accuracy, miss rate, and recall of ternary models and comparing those values between the All-feature model (ALL), most frequent features model (MFF), and significant features model (SF). This is to identify the potential for a better performing model with fewer features.

### 3.3 Accuracy

The differences in accuracy between the three approaches are barely visible when looking at the median (ALL: 75.08%, MFF: 78.20%, SF: 73.95%), but even greater when comparing the 75th percentile (ALL: 80.989%, MFF: 85.44%, SF: 79.25%, see Fig 7). All models show a wider range of accuracy values which means these models might overfit more on some runs and underfit on others. The lower adjacent of all models is higher than chance level (ALL: 53.46%, MFF: 52.93% and SF:52.41%), which means all models perform better as guessing. The chance level for 3 classes is 33.33%. A system that only guesses the correct class usually ends up with an accuracy of about 33.33%. Although not in each run, on average all models show a much better performance. Even the worst classification is over 20% higher than chance level. A successful performance for classification expertise in machine learning models is typically when their average accuracy is between 70% and 80%. A statement about the performance of a model with lower than 70% accuracy depends on the task and how much data is available. Sometimes there are only a few people in the world who are truly considered to be experts. As the accuracy is a rough performance metric that only provides information about the number of correct predictions (true positives and true negatives), we offer a more detailed look into the performance of theses methods by comparing the miss rates of single approaches.

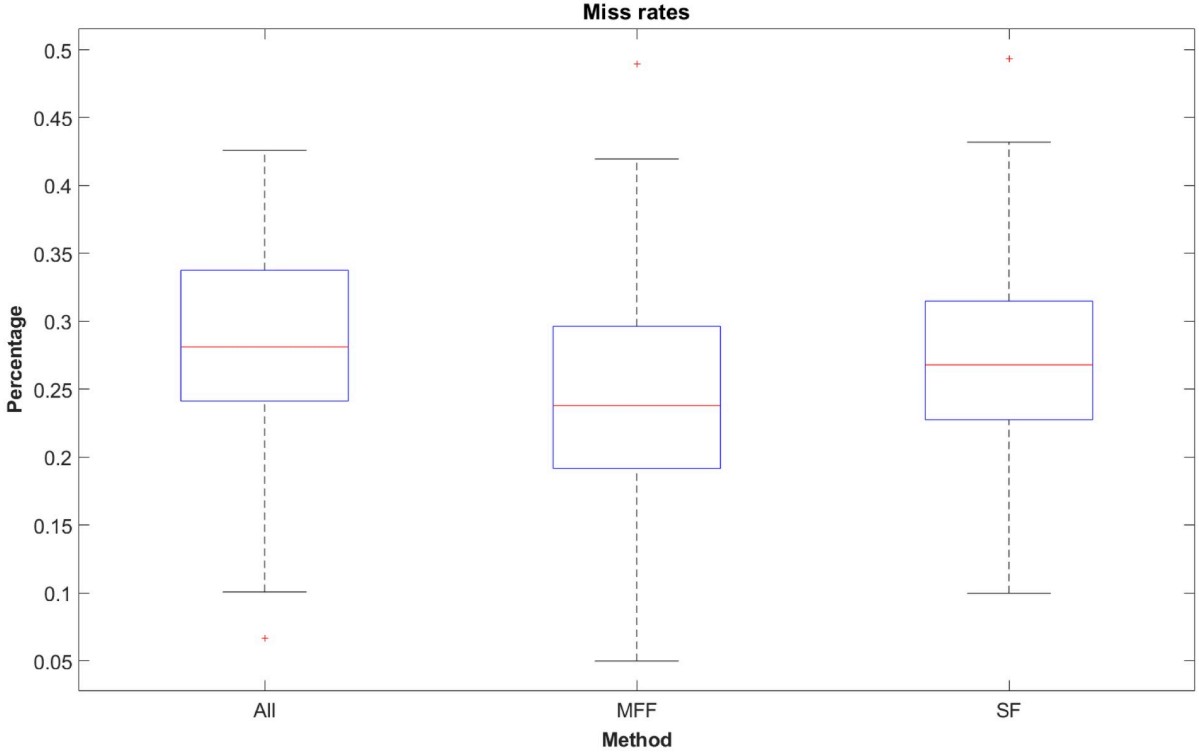

**Fig 8. Miss rates of ternary methods.**

### 3.4 Miss rate

The miss rate is a metric that measures the rate of wrongly classified samples belonging to class x, but predicted to belong to class y. The models are better at predicting the membership of samples belonging to expert and intermediate classes than to the novice class. This results in miss rates that are only little lower than chance level when looking at the median miss rates (All: 28.12%, MFF: 23.81% and SF: 26.80%, see Fig 8). The upper adjacent shows a high range of miss rates reaching even values of over 43.19% for the SF-model. The MFF-model has the lowest median miss rate of all three methods with a miss rate of 41.96%.

### 3.5 Recall

Recall provides information about the rate of predicted samples belonging to class x in relation to the number of samples that really belong to class x. All three models have a median recall of over 70% (as can be seen on Fig 9). In the ternary case, chance level is at 33.33% which means all models have a recall over two times higher than chance level as the lower adjacent of all three models is higher than 33.33%. The MFF-model median is the highest at 76.18% followed by the SF-model at 73.194% and the ALL-model at 71.87%. Again, the MFF-model has the best performance values of all three methods.

### 3.6 Most frequent features

The most frequent features in 100 runs are summarized in Table 6. Only the minimum of the saccade duration has $p > 0.011$. This means the differences are not statistically significant. All

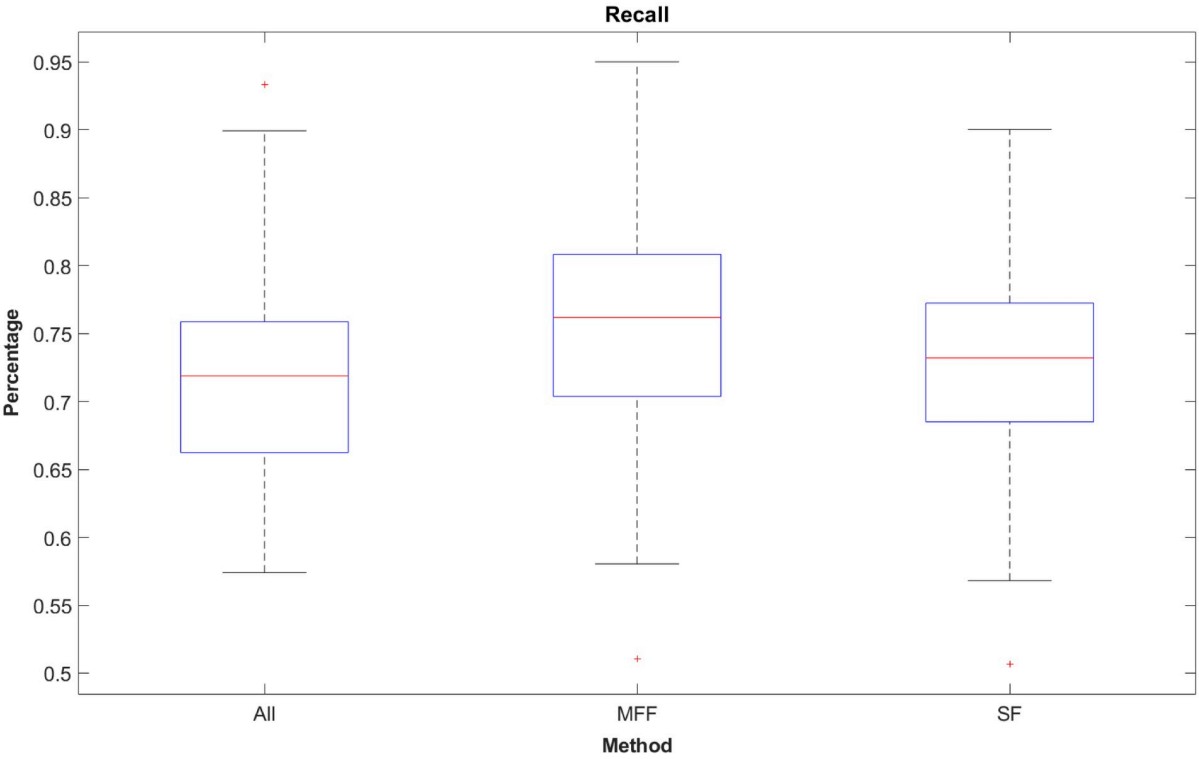

**Fig 9. Recall values of ternary methods.**

other features show significant differences, indicating that a Mann-Whitney-U-test discards the null hypothesis that there are no differences with $p < 0.011$ for each of the features.

## 4 Discussion

In this work, we have presented a diagnostic model to classify the eye movement features of soccer goalkeepers into expert, intermediate and novice classes. We further investigated how successfully the features provided by the diagnostic model resulted in explainable behaviour.

Our model has shown that eye movement features are highly informative and well suited to distinguish different expertise classes. Based on a support vector machine as a simple machine learning model, we were able to classify three different expertise groups at an average accuracy of 78.2% (compared to the baseline of 33.3% in a three-class classification problem), a quality

**Table 6. All most frequent features.**

| | | Most frequent features | | | | |
|---|---|---|---|---|---|---|
| **Features** | **derivation** | **novice** | **intermerdiate** | **expert** | **p-value** | **hypothesis discarded** |
| saccade duration (ms) | std. dev. | 38.869 | 59.726 | 35.548 | 3.33*e-08 | 1 |
| saccade duration (ms) | minimum | 26.514 | 26.127 | 25.019 | 0.242216408 | 0 |
| peak saccade deceleration ($°/s^2$) | std. dev. | 4166.262 | 4770.063 | 5492.287 | 2.49*e-18 | 1 |
| peak saccade velocity ($°/s$) | std. dev. | 129.294 | 136.529 | 157.740 | 6.19*e-07 | 1 |
| smooth pursuit dispersion (pixels) | average | 622.805 | 425.089 | 399.939 | 9.66*e-82 | 1 |
| smooth pursuit dispersion (pixels) | minimum | 185.437 | 168.320 | 336.016 | 5.44*e-12 | 1 |
| smooth pursuit dispersion (pixels) | maximum | 1085.903 | 694.370 | 505.031 | 1.52*e-81 | 1 |

result for current machine learning techniques. As the performance values differ, the real-world application will have to be further evaluated with larger subject groups.

A closer look at the classification results reveals that our model can successfully distinguish between experts and intermediates. This is due to the fact that experts and intermediates have already proven their ability by playing in the higher leagues. Thus, there is a ground truth for these classes. A limitation of the classification model is the novice group. Since our novice group consists of participants with no regular training or involvement in competitions, novices can be equally talented players, as revealed through their gaze behaviour, who have yet to prove their ability in a competition. This assumption is especially evident in the false negative rate of 1.6% and the false positive rate of 18.6% from the binary model. This means that 18.6% of novice samples are classified as intermediate samples, but only 1.6% of the intermediate samples are classified as novice. As is typical in expertise research, a portion of low performers (novices) can also be found in higher classes. Our models confirm that the correct classification of novices is considerably more difficult than other classes because there is, to date, no objective ground truth. Despite this limitation, our model achieved a very good average accuracy of 78.2%. Most likely, a model with more subjects and a finer graduation of the novice class would achieve better results. Machine learning models are data-driven and therefore learn more from more data. However, the number of elite youth goalkeepers in Germany who can provide samples for the expert class is limited. Out of 56 in total, we only collected data from 12 in our study. Defining a more robust ground truth for participants classified as novices is an important step for future models. Since the current model fundamentally does not downgrade participants with higher expertise to a lower class, it can still be used as a diagnostic model. As aforementioned, the false positive rate shows that only some novices with limited experience perform better than others and can, therefore, be classified into a higher class. Namely, their gaze behaviour is closer to that of intermediates than it is to typical novices.

By examining the individual eye movement features in more detail, we have shown that a subset of features is sufficient to create a solid classification. We also found, however, that the difference in eye movement behaviour between the individual groups is difficult to interpret. We only investigated the most frequent features because they built the best performing model. The differences are noticeable, but hard to interpret due to the fact that there are no simple characteristics behind these features.

There are indications that experts (std. dev. 35.54 ms) and novices (std. dev. 38.86 ms) have a more homogeneous saccade behaviour when compared to intermediates (std. dev. 59.72 ms). The lengths of the saccades differ less. However, it would be a fallacy to attribute the same viewing behavior to novices and experts due to the similar standard deviation and minimum duration of the saccades (novice: 26 ms, intermediate: 25 ms, expert: 25 ms). It is clear that both groups have saccades of a similar length, but the novices have similarly long saccades and the experts similarly short saccades. Conversely, this means that the experts may have longer fixations than novices and intermediates. These findings are in line with Mann et al. [20] who show that experts are over-represented in fewer, but longer fixations. Their visual strategy is often based on longer fixations to avoid saccadic suppression (which might lead to information loss). In our statistics, the duration of fixations did not differ significantly between the three groups which is in line with the findings of Klostermann et al. [11]. This may be based on the division of fixation values into short fixations and smooth pursuits. Differences could also stem from the age difference between single groups (see Table 1). With the current data, it is difficult to determine the reasoning behind the differences with any certainty.

Further differences between groups can be found in the maximum peak deceleration of the saccades. There is a continuous increase in the maximum deceleration speed of the novices' saccades ($4166.262°/s^2$) to intermediates($4770.063°/s^2$) when compared to experts ($5492.287°/$

$s^2$). This is in line with the findings of Zwierko et al. [42] who found that deceleration behaviour can be inferred from different expertise classes.

One observation made by the experimenter during the study was that novices often follow the ball with their gaze for a long time. This behavior is less evident among experts. Experts only tend to look at the ball when it has just been passed or when they themselves are not in play. At such times, the ball can not change its path. This observation is supported by the values of the smooth pursuit dispersion. With 505.031 pixels maximum and 336 pixels minimum, experts have a very narrow window of smooth pursuit lengths. Essentially, the maximum smooth pursuit for experts (505.03 pixels) is less than half as long as for novices (1085.90 pixels) and the minimum smooth pursuit (expert: 399 pixels, intermediate 425 pixels, novices 622 pixels) is still 1/3 shorter than for novices. The intermediates are placed in the middle between the two groups. Again, the values are continuously decreasing. Based on the continuity of the average smooth pursuits that correlate negatively with the classes, as well as the maximum and standard deviation, it can be concluded that experts tend to make smooth pursuits that are more regular in length. One explanation for this could be that, in addition to the opponents and players, the ball, as an almost continuously moving object, attracts a high level of attention. In order to maintain a clear overview of the decision-making process, soccer players are taught the following behavior: shortly before the ball arrives at the pass goal, look at it. This is done until the ball is passed away. Since the path of the ball can only be changed by the player who is in possession of the ball and not in the middle of a pass, it is only necessary to follow the path of the ball at the beginning and end of the pass. In the meantime, players should scan the environment for changes to keep track of options in the field. This leads to short smooth pursuits around the ball before the end and at the beginning of each pass so that experts can appreciate the ball and follow the ball with similarly long smooth pursuits. On the other hand, as aforementioned, novices follow the ball's path almost continuously or, at least, very frequently. The characteristics of the smooth pursuit support this theory. The characteristics of smooth pursuits differ significantly from one another in the three groups with an average, minimum and maximum significant p-value of less than $1 * 10^-12$. The novices with 622.81 pixels make, on average, much longer smooth pursuits than the intermediates (525.09 pixels) and significantly more than the experts (399.93 pixels). With 185.44 pixels, the novices' shortest smooth pursuits are smaller than those of the intermediates (168.32 pixels) and the experts (336.01 pixels). The maximum values show a uniform behaviour. With 1085.9 pixels, the novices have the highest maximum values after the intermediates with 694.37 pixels and the experts with 505.03 pixels.

Although the standard deviation of the lengths of the smooth pursuits does not belong to the MF-features, clear differences can be seen here as well. The dispersion of smooth pursuits with 201.27 pixels scatters far more among the novices than among the intermediates (124.85 pixels) and experts (112.41 pixels). These findings lead us to believe that a stimuli oriented investigation of gaze distribution for expertise recognition might reveal even more pronounced differences, i.e correlation between ball movement and smooth pursuits.

## 4.1 Conclusion and implications

After the ternary classification of expertise, the next step should be the evaluation of a more robust classification model. As machine learning techniques are data-driven, adding more subjects to each group presumably leads to better results. A more sensible model with a finer grained gradation and the ability to classify participants into additional classes by predicting their class with greater nuance should also be considered. In future work, we plan to expand our data set to additional subjects in the current groups, add more nuanced classes, and add a physical response mode to infer speed and correctness in a standardized, controllable, and

objective manner, thus increasing immersion. However, a fully interactive mode will only be possible when CGI can provide cost-efficient, high quality environments. Another step is to focus on the research of person-specific, gaze-based expertise weakness detection. As soon as a robust model is achieved, we plan to integrate the model into an online diagnostic system. To use the model online, the gaze signal can be drawn directly online at 250 Hz from the eye tracker by using the vendor's API. Using a multi-threaded system, data preparation and feature calculation can be performed directly online in parallel with data collection. Only the higher level features (e.g. std. deviations) need to be computed when the trial ends and fed as feature vector to the already trained model in order to estimate the class of the current trial. As prediction is completed by solving a function, the prediction result is supposed to be available a few moments after the trial ends. This is necessary as the prediction is the input for the adaption of the training. This work will be implemented in an online system for real-time gaze-based expertise detection in virtual reality systems with an automatic input for the presentation device to ensure dynamic manipulation of a scene's difficulty. With a prototype running in VR, we are planning to expand the system to be used in-situ with augmented reality-glasses (AR). This may even further pronounce differences and lead to better classifications. By mapping expertise on a larger number of classes, a more sensible model would allow for the dynamic manipulation of difficulty level in a virtual training system exercise or game level. Next to a training system for athletes and other professional groups, the difficulty level of a VR game can be dynamically adjusted based on the user's gaze behavior. We are, however, aware that the small sample size restricts potential conclusions and could lead to contentious results. Another limitation of this work is the restriction presented by head movements unrelated to eye movement features and the absence of a detailed smooth pursuit detection algorithm. Therefore, in our future work we will implement an appropriate event calculation method i.e. based on the work of Agtzidis et al. [43]. This work does, however, strengthen the assumption that there are differences in the gaze behavior of experts, intermediates and novices and these differences can be ascertained through the methods discussed.

Using machine learning techniques on eye tracking data captured in a photo-realistic environment as displayed through virtual reality glasses is a potential first step towards the development of a virtual reality training system (VRTS). Objective expertise identification and classification can lead to the adaptive and personalized design of such systems because these measures have the ability to define certain states in a training system. For example, a VRTS that can be used at home and, based on its objective and algorithmic kind, allow for self-training. The choice of difficulty can be adapted based on the expertise of the user. For highly skilled users, the level of difficulty can be raised by pointing out fewer cues or showing more crowded, faster, and dynamic scenes that increase the pressure placed on decisions. With enough data, it is also possible to adapt the training level based on personal deficiencies discovered during expertise identification in a diagnostic system. This can result in a system that knows a user's personal and perceptual weak spots in order to provide personalized cognitive trainings (e.g. different kinds of assistance like marking options, timing head movements, showing visual and auditory cues). Such a system is also potentially applicable in AR. The findings on the photo-realistic VR setup can be used in AR settings (i.e. in-situ). For uses like AR trainings, that are meant to enhance physical trainings, fundamental findings must be rooted in real gaze signals. As a second step, training systems can be developed based on diagnostic findings. This is, in addition to physical training, especially relevant as there is increasing research on forms of perceptual-cognitive training going on. [44–47].

## Supporting information

**S1 Video. Example video of experimental VR environment.** The video first shows the view at the inside of the sphere. With the head of the participant and the field of view. There is an example gaze signal jumping between players. A real stimulus is played on the inside of the sphere. Later the camera zooms out of the sphere to show how the video is projected on its inside.
(MP4)

## Author Contributions

**Conceptualization:** Benedikt W. Hosp, Oliver Höner.

**Data curation:** Florian Schultz.

**Formal analysis:** Benedikt W. Hosp.

**Funding acquisition:** Florian Schultz, Oliver Höner.

**Investigation:** Benedikt W. Hosp.

**Methodology:** Benedikt W. Hosp, Florian Schultz.

**Project administration:** Florian Schultz, Oliver Höner, Enkelejda Kasneci.

**Resources:** Florian Schultz.

**Software:** Benedikt W. Hosp.

**Supervision:** Florian Schultz, Oliver Höner, Enkelejda Kasneci.

**Validation:** Benedikt W. Hosp.

**Writing – original draft:** Benedikt W. Hosp.

**Writing – review & editing:** Benedikt W. Hosp, Florian Schultz, Oliver Höner, Enkelejda Kasneci.

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
