## [Decision Letter · Decision Letter 0]

24 Nov 2020

PONE-D-20-29324

Eye movement feature classification for soccer expertise identification in virtual reality

PLOS ONE

Dear Dr. Hosp,

Thank you for submitting your manuscript to PLOS ONE. After careful consideration, we feel that it has merit but does not fully meet PLOS ONE’s publication criteria as it currently stands. Therefore, we invite you to submit a revised version of the manuscript that addresses the points raised during the review process.

Both reviewers agree that the paper has merit but that it currently needs major revisions. These relate to the structure of the article, the lack of clarity and purpose of the work and significant issues with the English language used throughout. These issues are fully explained by the reviewers in their comments below.

We look forward to receiving your revised manuscript.

Kind regards,

Greg Wood, PhD

Academic Editor

PLOS ONE

Journal Requirements:

2. "PLOS ONE does not copy edit accepted manuscripts. Please proofread for typos and grammar as well as for the use of commas instead of decimal points.

"This research was supported by the German Football Association (DFB)."

"The author(s) received no specific funding for this work"

6. Please ensure that you refer to Figures 2, 5, 7, 8, 9 in your text as, if accepted, production will need these references to link the reader to the figures.

Reviewers' comments:

Reviewer's Responses to Questions

**Comments to the Author**

1. Is the manuscript technically sound, and do the data support the conclusions?

Reviewer #1: No

Reviewer #2: Partly

2. Has the statistical analysis been performed appropriately and rigorously? 

Reviewer #1: I Don't Know

Reviewer #2: Yes

3. Have the authors made all data underlying the findings in their manuscript fully available?

Reviewer #1: No

Reviewer #2: Yes

4. Is the manuscript presented in an intelligible fashion and written in standard English?

Reviewer #1: No

Reviewer #2: No

5. Review Comments to the Author

Reviewer #1: To the author:

I would like to thank the authors for the opportunity to review this work. I found it very interesting and commend the researchers on their efforts. However, I do feel there are a number of issues, which are in general linked to a lack of clarity about the purpose. See my comments below. I have also pasted my comments to the editor underneath in the interest of transparency.

-Dr David Harris, University of Exeter.

Abstract

Typo: All scenes WERE

General comments:

One of my concerns with this study is that just because we can use this type of machine learning approach, it doesn’t mean we should. There is certainly a place for it, but I don’t think the authors did a good job of motivating its use. It doesn’t tell us anything about what perceptual expertise actually is. Lots of variables are thrown in an algorithm but we don’t learn anything about what experts do differently. So what is the purpose? The purpose can’t actually be identifying experts, because we would fare much better by just asking how often they play soccer.

The rationale for the study seems to be 1) that VR is better than screen-based videos for studying sporting expertise, and 2) that machine learning is an interesting analysis technique. While I do not disagree with either point, both are methods for answering research questions, but neither is a research question in itself. Consequently I think the introduction needs to do a better job of motivating the study. At the end of the introduction I was still unsure what question you are answering and what we are going to learn from this work.

While the authors discuss the benefits of VR for making perceptual-cognitive research better by enabling more realistic eye movements, their approach sort of misses the point of why this is useful. The authors analyse eye movements in a way that they are completely disconnected from what the participants are viewing. Eye movements are meaningful in relation to the stimuli that the person is viewing (unless they are being used to infer a general state like anxiety, which is not the case here). Lets say, for instance, you find that experts are showing larger saccades. That finding is pretty meaningless when it is out of the context of what they were viewing. Unless the eye movements are coupled to the scene it is very hard to interpret what any of them mean or what they tell us about expertise. I just feel that this approach misses the benefit of doing this work in VR.

It seems like this hasn’t been proofread very well which is frustrating. There are full stops in the wrong place, missing capital letters, misspellings and parentheses used incorrectly. Things that a spell checker would find. I started to try noting these down but stopped because I felt that a proper proofread by the authors could have caught them.

Specific comments:

L4 – While it is common, I think some researchers would argue that video-based research is not optimal.

L32 – typo

L39 – I think this is a really key point - the assumption that gaze behaviour will be the same in VR. Which I am pretty confident has not been properly tested. I am in agreement that VR is a good way to examine perceptual-cognitive skills but the evidence that it elicits similar gaze behaviours is not there yet. Indeed, the extent to which a VR environment elicits realistic gaze behaviour is likely to be environment specific, and would be referred to as the ‘psychological fidelity’ of the environment.

Both these papers discuss this issue:

-Gray, R. (2019). Virtual environments and their role in developing perceptual-cognitive skills in sports. Anticipation and decision making in sport, 342-358.

-Harris, D. J., Bird, J. M., Smart, A. P., Wilson, M. R., & Vine, S. J. (2020). A framework for the testing and validation of simulated environments in experimentation and training. Frontiers in Psychology, 11, 605.

L42 – as above. This really hasn’t been established.

L49 – I think ‘objective reproducible’ is debatable. Machine learning still depends on the decisions made by the researcher about how to treat the data. It not as assumption free as people would like to think.

L52 – full stop in the middle of the line.

L57 – perhaps you could explain more how it would lead to training. This section is a bit vague. Adjusting training difficulty to the level of expertise of the trainee is not something new. That’s how training always works.

L86 – typo.

L90 – the novice group seems quite disparate. It includes both soccer players and non-soccer players. Why was this choice made? Either low level players or non-players might make sense, but a mixture is an odd design choice.

L113 – I know the software calculates the fixations etc, but it would be useful to add what the parameters were. E.g. if you are interested in fixations what constitutes a fixation in this case, what constitutes a saccade etc. How was smooth pursuit calculated? I have used BeGaze software but do not recall it calculating smooth pursuit.

I also think there needs to be some discussion of what all these metrics mean. There must have been a rationale for including them and it would be useful to explain to the reader how to interpret the measures. For instance what does average saccade deceleration show about the perceptual behaviour of the performer? What does it mean in the context of this task? Otherwise they are just numbers.

L32 – please report how many trials (and %) were lost. And were any participants totally removed? (I do see that some of this is reported below, but an overall figure would be useful)

‘we only used trials, that we consider as valid’ sounds disconcertingly vague. Were trials removed for any other reason than the proprietary tracking ratio? If so what?

L138 – invalid in what way?

L151 – what do you mean by samples 7, 8, 14-16, 18-20? Are these participants? Or parts of every trial? Either way it seems concerning that these were lost.

There is lots of details on what an invalid saccade amplitude value is, but you haven’t even described how a saccade was identified as a saccade. Similarly there is little detail on how fixations were treated, filtered etc.

L184 – how can this work when there were only 8 intermediates?

I am not familiar with how to gauge power for machine learning models but I think some discussion of why this number of participants and trials was appropriate is needed. 8 in a group would generally be considered quite low.

L220 – bracket typo.

L264 – typo

L295 – how are these p values corrected for the multitude of comparisons that are made?

L298 – this is a real over-interpretation of the standard deviation values when the value for the intermediate gorup might just be higher because the sample is smaller (indeed that is how a standard deviation works).

Fig 7 – it would be useful to put some detail in the figure legend about what is plotted. Like what are those error bars?

Also the prediction interval seems to be pretty wide; its somewhere between perfect and 50%. Is this a reflection of the limited size of the data set?

Your results focus on the comparison the different models, which all provide a fairly similar level of prediction, but I think it would be more useful to discuss what this means in real terms. Whether one model is better than another is really much less relevant that considering what a good level of prediction is in this context? What would be a good/acceptable/poor level of accuracy? What level of accuracy is needed for the type of training you discuss? With these types of statistical approaches it is very easy to get abstracted from the actual context (one of the contextual issues being that I can get 100% accuracy by just asking them).

L298 – this could just be a function of your definition of ‘intermediate’. What ages are the intermediates? (More detail needed in the participant section) because it sounds like they might be adults, in which case they might have more soccer experience than the experts. Similarly some of the novices could have had plenty of experience, but just be less good.

L307 – no they don’t have longer to process information. They use a visual strategy of longer fixations because it is more effective for extracting information because of saccadic suppression.

It is surprising that well established features of perceptual-cognitive expertise, like longer fixation durations, were not found. Might be worth discussing.

L328 – I don’t see how having a faster maximum saccade (which is not actually the same as faster saccades as you state) means that experts adapt themselves to the situation well. They almost certainly do, but it’s a huge leap to say that this data shows that.

L331 – where has this data come from? Is this just an observation of the experimenter during testing? This kind of data that is linked to actual stimuli is much more meaningful for understanding expertise than, say, standard deviation of peak saccade deceleration.

L342 – such a setup as what?

It seems odd to start a discussion with the limitations. It would be more conventional to restate your research question, your hypotheses and then say whether the results supported them or not.

L349 to 354 – I don’t see how the findings have shown this in any way. You have not shown that the gaze behaviour was similar to natural gaze behaviour, you haven’t even measured natural gaze behaviour, so how can you conclude this?

L361 – again this is a jump. How do you know its structured without relating the scan paths back to the visual environment? The experts could have been scanning all over the place.

L366 – I really like this aim for training based on personalised gaze behaviour.

To the editor:

I was asked to review the manuscript ‘Eye movement feature classification for soccer expertise identification in virtual reality’. In this study, the authors use a machine learning approach to classify the gaze behaviour of expert, intermediate and novice soccer goalkeepers when watching 360 video of game footage. They report a model with ~75% accuracy in classifying participants as novice, intermediate or expert. They also find some features of saccadic eye movements that are linked to expertise.

While the work is sound (although I cannot comment on some of the specifics of the machine learning approach) I am unsure about the value or purpose of the work. The authors are really focused on the method (VR and machine learning) but do not really make it clear why this work needed to be done. This is reflected in the muddled discussion which does not really conclude anything related to the current findings, but makes some statements that are unsupported by the data. Essentially I anm unsure what we can learn from this work. The English use is also quite poor (and has not been proofread properly) and the sample size is pretty small. Perhaps it could be revised, but I cannot recommend for publication in the current state.

Reviewer #2: The authors go far to describe how they have built up the system and they present many interesting details about the ML aspect of their proposal. Given that sport science is still very new to these types of systems, the authors offer some potentially very useful perspectives that might be of interest to both researchers, practitioners and technology providers in sport.

With that said, there are some issues that need to be addressed before this article is worthy of publication.

I have made comments according to the criteria for publication below.

1. The study presents the results of primary scientific research.

- The study presents some results of primary research; however, the quality of this research appears to be somewhat limited in sample size. Also, due to a lacking literature research, their novelty regarding the sports science field is rather unclear (the novelty is there, I just want them to show it more clearly, more on this in my following comment).

-The literature review is rather limited. Regarding the VR aspect, the merit of choosing specifically that technology above others is somewhat explained. However, is rather surprising to see that the utilization of VR in other sports fields such as tennis is introduced, while the soccer specific applications are not. Recent work by Aksum focusing on eye fixations, as well as Wood or Rojas Ferrer in relation to VR applied to soccer training and skill assessment are some examples of work being pursued in this field. Also, researchers such as Jordet, Roca, Dicks, Vestberg, Savelsbergh, and McGuckian (and many more, but I acknowledge that not everyone should be given equal weight) - who all have done good work on visual perception in soccer in the last decade, are notably absent in the review.

2. Results reported have not been published elsewhere.

- As far as I am aware, the results have not been published elsewhere.

3. Experiments, statistics, and other analyses are performed to a high technical standard and are described in sufficient detail.

- The statistical analysis seems robust enough but lack some clarity in their interpretations due to bad overall readability of the paper (more on this in point 5).

-The subjects are divided in three levels of expertise. However, there is a significant difference between someone without any soccer experience and someone that plays in a low-ranked league. Being a bit more specific in their subject’s selection criteria would be good. Also, in relation to the participants I could not find the information about their age and gender. One could somewhat infer the ages of the expert group but not for the others. Is better to show the descriptive statistics of the participants if available.

-L108-L109 The criteria used to define "the option that has the highest probability to lead to a goal…" is unclear. If defined by experts then please provide some details or if it was based on related research please cite.

4. Conclusions are presented in an appropriate fashion and are supported by the data.

-The lack of a more substantial discussion section showing how the findings are relevant to the larger field of sport science with respect to both theory and other empirical studies on similar topics diminishes the importance of the substantial work done here. The discussion is really thin and should reflect more on theory and the field as a whole, not just the test of this specific system. As I also mentioned in point 1, a better related literature research in the Introduction section may also help in this regard.

5. The article is presented in an intelligible fashion and is written in standard English.

- Generally, this article needs substantial work in the organization of the paper and proofreading. As a reference some errors are outlined below, but as the paper stands, general improvement is needed.

- L12: "postulateed" should read "postulated"

- L20: “of a optimized” should read “of an optimized”

- L32: “virtzual” >> I am not clear if it lacks correct punctuation or if it is a subtitle. Furthermore, what is virtzual? I think it’s just a typo but if it is a concept or technology, then it should be properly cited and introduced.

- L64: “that are represent by…” should read “that are represented by...”

- L86: “The footage was player on the VR glasses” should read “The footage was played on the VR glasses” and errors like these are repeated throughout the paper.

- There are a number of confusing statements throughout the rest of the manuscript. While I appreciate the authors are likely writing in their second language, I recommend the authors have a native English speaker proof-read the manuscript for clarity.

-Finally, the structure of the paper seems a bit unconventional. I suggest the authors carefully structure the paper in order to ensure logical progression of ideas. I feel the paper would flow well in a more traditional sense (introduction, methods, results, discussion). Integrating the Project Description section into Methods, then subdividing the VR and ML components of the system description as dedicated sub-sections could be an alternative.

6. The research meets all applicable standards for the ethics of experimentation and research integrity.

- The research appears to meet ethical standards.

7. The article adheres to appropriate reporting guidelines and community standards for data availability.

- The article appears to meet appropriate data availability.

I wish the authors all the best with their research endeavors. The area of VR and Machine Learning in sports science is exciting, and there are possible contributions to be had from these authors.

6. PLOS authors have the option to publish the peer review history of their article (what does this mean?). If published, this will include your full peer review and any attached files.

Reviewer #1: **Yes: **Dr David Harris

Reviewer #2: No

---

## [Author Response · Author response to Decision Letter 0]

20 Feb 2021

Reviewer #2: Please see page 5.

We thank the reviewers for their valuable and constructive feedback, which have helped us to improve the manuscript. In the following we provide a point-wise response to the reviewer’s comments. Typos are corrected directly in the document (see “Revised Manuscript with Track Changes”). All changes are highlighted in the manuscript.

 General comments

1. One of my concerns with this study is that just because we can use this type of machine learning approach, it doesn’t mean we should. There is certainly a place for it, but I don’t think the authors did a good job of motivating its use.

2. It doesn’t tell us anything about what perceptual expertise actually is. Lots of variables are thrown in an algorithm but we don’t learn anything about what experts do differently. So what is the purpose?

3. The purpose can’t actually be identifying experts, because we would fare much better by just asking how often they play soccer. 

4. While the authors discuss the benefits of VR for making perceptual-cognitive research better by enabling more realistic eye movements, their approach sort of misses the point of why this is useful. 

5. The authors analyse eye movements in a way that they are completely disconnected from what the participants are viewing. Eye movements are meaningful in relation to the stimuli that the person is viewing (unless they are being used to infer a general state like anxiety, which is not the case here). Lets say, for instance, you find that experts are showing larger saccades. That finding is pretty meaningless when it is out of the context of what they were viewing. Unless the eye movements are coupled to the scene it is very hard to interpret what any of them mean or what they tell us about expertise. I just feel that this approach misses the benefit of doing this work in VR.

- 1) One major advantage of machine learning is its ability to work with big data. We added a paragraph to the introduction where we explain that we reached a point where we gain a lot of knowledge from high speed devices like eye trackers, but still use manual methods to infer high level knowledge from it. Machine learning is a consequent step since these techniques become stronger with more data. And therefore, allows experimenters to learn more about perception of athletes. Machine learning can also provide valuable insights on the importance of eye movement features to expertise. Since it is hard for human to describe a gaze behaviour by getting asked, machine learning is able to find latent feature and feature patterns, one can not simply ask. According to the reviewer’s feedback, we added a paragraph to motivate the usage of machine learning in our approach. We hope it is much clearer now. 

- 2) One disadvantage of machine learning is the explainability. Often, machine learning methods find patterns that are not that obvious for human to understand. But in this work we do not focus on the understanding of differences between expertise groups, since we first want to find out whether expertise can be classified with these methods. We use a lot of variables to have a higher chance of finding a subset that predicts expertise with a small amount of features. Not to find a subset that explains the differences, but allows a machine learning based classification of expertise. In a further work we want to focus on explainable differences.

- 3) An expert goalkeeper does not need to have a high expertise in perceptual skills. Some only have good goalkeeper skills. Eye tracking reveals subconscious behaviour and therefore allows to investigate on non-obvious behaviour. Most players cannot properly tell where they are looking and why as subconscious processes are involved. With the combination of eye tracking and machine learning we can work on much deeper levels of perception. We can not ask athletes where did you look when, how long do you follow the ball, how important is the length of your smooth pursuit. This all happens in subconscious processes. For a training system we need to find the fine nuances in their gaze.

4) Stimuli in VR lead to an increase in external validity. This increases the probability that the participants will perform a natural gaze behaviour. Only in this way it is possible to test whether the level of expertise can be recognised on the basis of eye movements. We need realistic eye movements as at some point, since the motivation of our work is also to develop a diagnosis system, that can be used at home especially by young athletes to diagnose their expertise. These findings will provide a basis to build a training system in a next step to enhance perceptual expertise of young talents in a home training system.

- 5) The experimental control of this study allows us to infer the expertise of these participants in a stimulus unrelated manner, but close to natural environment. You are right, eye movements are meaningful in relation to stimulus, yes. But as every participant saw the same videos we can abstract their behaviour. We pointed out, that a stimulus-oriented investigation might reveal more pronounced differences. But as every participant saw the same videos we can abstract their behaviour. Our samples are average values over one video trial. As we have a lot of features that describe the gaze behaviour, the differences can be reflected in more than one feature. We want to condense the amount of features to know how we can abstract the expertise based on average features of one video into expertise classes. 

We do not think the findings are meaningless. Because we assume that participants act always more or less the same when the task is decision-making under high pressure. But you are right: Without the relation to the stimulus, it is really hard to interpret the features. As we stated, the smooth pursuits need to be investigated further, since we might have found a correlation between the movement of the players and the smooth pursuits. For the diagnostic system it is less important for us to know how the differences in gaze behaviour are, but more whether the differences can be used to differentiate participants into expertise classes.

Specific comments:

6. L39 – I think this is a really key point - the assumption that gaze behaviour will be the same in VR. Which I am pretty confident has not been properly tested. I am in agreement that VR is a good way to examine perceptual-cognitive skills but the evidence that it elicits similar gaze behaviours is not there yet. Indeed, the extent to which a VR environment elicits realistic gaze behaviour is likely to be environment specific, and would be referred to as the ‘psychological fidelity’ of the environment.

Both these papers discuss this issue:

-Gray, R. (2019). Virtual environments and their role in developing perceptual-cognitive skills in sports. Anticipation and decision making in sport, 342-358.

-Harris, D. J., Bird, J. M., Smart, A. P., Wilson, M. R., & Vine, S. J. (2020). A framework for the testing and validation of simulated environments in experimentation and training. Frontiers in Psychology, 11, 605.

- Changed and added psychological fidelity. Please see p.3, r. 42 ff

7. L49 – I think ‘objective reproducible’ is debatable. Machine learning still depends on the decisions made by the researcher about how to treat the data. It not as assumption free as people would like to think.

- Partly, this is true. But as long as the parameter set of the model is published too (as we did), it is easy to reproduce the model. The paper needs to describe the way parameters were set. Usually “why” cannot be answered, as this is mostly an empirical search or grid search.

8. L57 – perhaps you could explain more how it would lead to training. This section is a bit vague. Adjusting training difficulty to the level of expertise of the trainee is not something new. That’s how training always works.

- We moved the description of the training system to implications section and added a more detailed description on how this can lead to a training system. 

p. 26 r. 533 ff

9. L90 – the novice group seems quite disparate. It includes both soccer players and non-soccer players. Why was this choice made? Either low level players or non-players might make sense, but a mixture is an odd design choice.

- That’s a good point. Actually, the novices had once played either in a youth league or never at all. This means, the novice group contains participants who have no training on a regular basis and are not participating in competitions. We added a table containing attributes of the single groups. See p. 7, table 1

10. L113 – I know the software calculates the fixations etc, but it would be useful to add what the parameters were. E.g. if you are interested in fixations what constitutes a fixation in this case, what constitutes a saccade etc. How was smooth pursuit calculated? I have used BeGaze software but do not recall it calculating smooth pursuit.

- We added more details on this procedure in p. 7 and 8. The software first calculates the saccades based on the peak threshold, which means the minimum saccade duration (in ms) varies and is set dependent on the peak threshold default value of 40°/s. In a second step the software calculates the fixations. Samples are considered to belong to a fixation, when they are between a saccade or blink. With a minimum fixation duration of 50 ms we reject all fixations below this threshold. As there is no generally applicable method for detection of smooth pursuits, this kind of event is included and encoded as fixations with longer duration and longer dispersion in the fixations, too. We marked fixations with a fixation dispersion of more than 100 px as smooth pursuits. By doing this, we split fixations into normal length fixations and long fixations which we consider to be and call smooth pursuits. This threshold is an empirical value based on the sizes of the players as main stimuli in the video. The following section describes the steps that are necessary to train a model based on these eye movement features

11. I also think there needs to be some discussion of what all these metrics mean. There must have been a rationale for including them and it would be useful to explain to the reader how to interpret the measures. For instance what does average saccade deceleration show about the perceptual behaviour of the performer? What does it mean in the context of this task? Otherwise they are just numbers.

- We added a paragraph in the manuscript where we talk about the different metrics. Explaining all features would be too much for the paper. We took only the most frequent features and explained them in a more detailed way. As we start with a brute force method and find the important features during the training, we only consider the most frequent features. These features have the highest importance for the model to predict. Please see p. 22 for a discussion.

12. L132 – please report how many trials (and %) were lost. And were any participants totally removed? (I do see that some of this is reported below, but an overall figure would be useful)

- No participants were removed. We added a finer description and a table on p. 10. See table 2 for an overview. 

13. L151 – what do you mean by samples 7, 8, 14-16, 18-20? Are these participants? Or parts of every trial? Either way it seems concerning that these were lost.

There is lots of details on what an invalid saccade amplitude value is, but you haven’t even described how a saccade was identified as a saccade. Similarly there is little detail on how fixations were treated, filtered etc.

- Please see Method chapter for updated description about saccades and fixations (p. 4, r. 82 ff . and p.8, r. 181ff).

- These samples are gaze signal samples (see p.12, Fig. 3 caption)

14. L184 – how can this work when there were only 8 intermediates?

- The number 8 is correct. Since we only have 10 intermediates we use 8/13 novices, 8/10 intermediates and 8/12 experts. There was a typo at another point saying there were only 8 intermediates. So we have 2 intermediate left for validation, p.13 r. 266 ff.

15. I am not familiar with how to gauge power for machine learning models but I think some discussion of why this number of participants and trials was appropriate is needed. 8 in a group would generally be considered quite low.

- The chance level for 3 classes is 33.33 %. A system that would only guess the correct class would usually end up with an accuracy of about 33.33 %. All three models show a much better performance. Not in each run, but on average. But even the worst classification is over 20 % higher than chance level. Machine learning models for classification expertise are usually considered as good performing, when their average accuracy is between 70 % and 80 %. A statement about the performance of a model with lower than 70 % accuracy depends on the task and how much data is available. Sometimes there are only a few people in the whole world, considered as experts. 

- We are aware that the sample size might restrict the conclusions, that can be drawn but the number of experts is strictly limited. We were very lucky to get 12 experts in our study. Please note that our method has no problems in detecting experts. The problem is more on classifying novices (mentioned in the discussion).

- 

16. L295 – how are these p values corrected for the multitude of comparisons that are made?

- The desired alpha level is 0.05 (5%). We corrected this level with the Bonferroni method by dividing the desired alpha level by the number of tests (46). This results in a significance level of 0.0011 (0.11%). 

17. L298 – this is a real over-interpretation of the standard deviation values when the value for the intermediate group might just be higher because the sample is smaller (indeed that is how a standard deviation works).

- We stated on p. 24, r. 444 ff. that it would be wrong to draw a conclusion about the standard deviation and added some other possible sources.

18. Fig 7 – it would be useful to put some detail in the figure legend about what is plotted. Like what are those error bars?

- Added a caption to figure 7.

19. Also the prediction interval seems to be pretty wide; its somewhere between perfect and 50%. Is this a reflection of the limited size of the data set? 

– No it is no reflection of the dataset. On a binary decision, 50% is chance level. 50% would be the 

 accuracy, if the model would guess. Everything better than this, is considered acceptable. 

 Everything over 70% is considered good. We added this discussion to the manuscript as well. 

 Please see p. 22, r. 404 ff. 

20. Your results focus on the comparison the different models, which all provide a fairly similar level of prediction, but I think it would be more useful to discuss what this means in real terms. Whether one model is better than another is really much less relevant that considering what a good level of prediction is in this context? What would be a good/acceptable/poor level of accuracy? What level of accuracy is needed for the type of training you discuss? With these types of statistical approaches it is very easy to get abstracted from the actual context (one of the contextual issues being that I can get 100% accuracy by just asking them).

- The idea of the different models is to understand that we can build a model, which is as good as 

the all feature model with much less features. As we can use less features, we get the information of which features of the gaze behaviour are important for a SVM to predict well. We added a

paragraph to explain this part to page p. 16, r. 310 ff. Another point to consider is the 

following: a goalkeeper expert (who is classified as expert because of the league he is playing in),

does not need to have a very good gaze behaviour. Maybe he is playing in this high league because

he is really good in keeping the ball off the goal but has problems in deciding how to continue the 

game after a return pass, because of a bad gaze behaviour. Our diagnosis system could discover 

such athlete. The integration of the goalkeeper into the game as a player not just as a goalkeeper is 

becoming more and more important these days, which makes it necessary to understand their gaze 

behaviour and build a training system to overcome these problems. Still the main task remains

keeping the ball off the goal.

21. L298 – this could just be a function of your definition of ‘intermediate’. What ages are the intermediates? (More detail needed in the participant section) because it sounds like they might be adults, in which case they might have more soccer experience than the experts. Similarly some of the novices could have had plenty of experience, but just be less good.

- We added an overview of the participants. Please see p. 7, table 1. 

22. L307 – no they don’t have longer to process information. They use a visual strategy of longer fixations because it is more effective for extracting information because of saccadic suppression.

- Right. Corrected. Bad translation. P. 24, r. 448 ff.

23. It is surprising that well established features of perceptual-cognitive expertise, like longer fixation durations, were not found. Might be worth discussing.

- Actually, it is difficult to say, since we splitted the fixations and smooth pursuits by a certain threshold. If one has a look at all the fixation values alone, there might be the same findings. But since we focused on this split into small fixations and smooth pursuits, we can not properly confirm the common findings. We just mentioned it only indirectly by the saccades behavior (p. 24, r. 450 ff. ).

24. L328 – I don’t see how having a faster maximum saccade (which is not actually the same as faster saccades as you state) means that experts adapt themselves to the situation well. They almost certainly do, but it’s a huge leap to say that this data shows that.

- You are right. This conclusion is too far away. We removed this conclusion, since we can not

 proof it properly.

25. L331 – where has this data come from? Is this just an observation of the experimenter during testing? 

- Yes, as it is written on p. 24, r. 464 ff. We added some sentences about theoretical goals in 

 football training, that fits to this observation.

26. This kind of data that is linked to actual stimuli is much more meaningful for understanding expertise than, say, standard deviation of peak saccade deceleration.

- We did that in a further study with the same stimuli and will compare the results between a stimulus oriented and stimulus-independent approach. We also added a paragraph to state that it might be better to go with stimulus-oriented p. 25, r. 501 ff.

27. It seems odd to start a discussion with the limitations. It would be more conventional to restate your research question, your hypotheses and then say whether the results supported them or not.

- Thank you. We rearranged the discussion and conclusion sections.

28. L349 to 354 – I don’t see how the findings have shown this in any way. You have not shown that the

gaze behaviour was similar to natural gaze behaviour, you haven’t even measured natural gaze 

behaviour, so how can you conclude this?

- Correct. We did not measure natural gaze behaviour. We wanted to point out that we used 

 VR-glasses and participants told us they feel really immersed into the scene. 

 

Reviewer 2:

Dear Reviewer,

Thank you for your valuable comments on our manuscript. We appreciate your help to improve our work. In the following we want to address your concerns. Typos are corrected in the manuscript directly. Please have a look at “Revised Manuscript with Track Changes”.

29. The study presents the results of primary scientific research. The study presents some results of primary research; however, the quality of this research appears to be somewhat limited in sample size.

- We are aware that the sample size might restrict the conclusions, that can be drawn but experts are busy people, so DFB could give us only enough time to record 12 of them. We were very lucky to get 12 out of overall 56 experts in our study. Also, please note that our method has no problems in detecting experts. The problem is more on classifying novices (mentioned in the discussion).

30. Also, due to a lacking literature research, their novelty regarding the sports science field is 

rather unclear (the novelty is there, I just want them to show it more clearly, more on this in 

my following comment). The literature review is rather limited. Regarding the VR aspect, the merit of 

choosing specifically that technology above others is somewhat explained. However, is rather surprising

to see that the utilization of VR in other sports fields such as tennis is introduced, while the 

soccer specific applications are not.

 - We added some of your mentioned researchers to the introduction to demarcate ourselves.

31. The subjects are divided in three levels of expertise. However, there is a significant difference between someone without any soccer experience and someone that plays in a low-ranked league. Being a bit more specific in their subject’s selection criteria would be good. Also, in relation to the participants I could not find the information about their age and gender. One could somewhat infer the ages of the expert group but not for the others. Is better to show the descriptive statistics of the participants if available.

- Added more details about subjects. Added table containing age, training/week and active years (see p. 6).

32. L108-L109 The criteria used to define "the option that has the highest probability to lead to a goal…" is unclear. If defined by experts then please provide some details or if it was based on related research please cite. 

- This was a wrong statement. We added an explanation.

33. Generally, this article needs substantial work in the organization of the paper and proofreading. As a reference some errors are outlined below, but as the paper stands, general improvement is needed.

- We organized the paper differently now by using a more common structure and we have a native speaker proof reading it.

34. Finally, the structure of the paper seems a bit unconventional. I suggest the authors carefully structure the paper in order to ensure logical progression of ideas. I feel the paper would flow well in a more traditional sense (introduction, methods, results, discussion). Integrating the Project Description section into Methods, then subdividing the VR and ML components of the system description as dedicated sub-sections could be an alternative.

- Thank you for this feedback. We changed the structure of the paper and hope it is now much clearer.

---

## [Decision Letter · Decision Letter 1]

7 Apr 2021

PONE-D-20-29324R1

Soccer Goalkeeper Expertise Identification based on Eye Movements

PLOS ONE

Dear Dr. Hosp,

Thank you for submitting your manuscript to PLOS ONE. After careful consideration, we feel that it has merit but does not fully meet PLOS ONE’s publication criteria as it currently stands. Therefore, we invite you to submit a revised version of the manuscript that addresses the points raised during the review process.

We look forward to receiving your revised manuscript.

Kind regards,

Greg Wood, PhD

Academic Editor

PLOS ONE

Journal Requirements:

Reviewers' comments:

Reviewer's Responses to Questions

**Comments to the Author**

1. If the authors have adequately addressed your comments raised in a previous round of review and you feel that this manuscript is now acceptable for publication, you may indicate that here to bypass the “Comments to the Author” section, enter your conflict of interest statement in the “Confidential to Editor” section, and submit your "Accept" recommendation.

Reviewer #1: (No Response)

2. Is the manuscript technically sound, and do the data support the conclusions?

Reviewer #1: Yes

3. Has the statistical analysis been performed appropriately and rigorously? 

Reviewer #1: Yes

4. Have the authors made all data underlying the findings in their manuscript fully available?

Reviewer #1: Yes

5. Is the manuscript presented in an intelligible fashion and written in standard English?

Reviewer #1: No

6. Review Comments to the Author

Reviewer #1: The authors have clearly made substantial efforts in revising the manuscript for which they should be commended. Their responses and the changes to the manuscript have clarified most of my concerns. I think some of my criticisms came from not fully appreciating what was (and more importantly was not) within the aims of this work. I think the paper is better for the revisions and presents some interesting findings. My only real remaining concern is with the writing/grammar, which still requires some work.

I just have a few other minor comments:

The use of references in the first few paragraphs of the introduction is quite infrequent, where one might expect background literature to be more regularly cited. Some more citations here about perceptual-cognitive expertise etc would be useful.

L13 – ‘a more highly developed perception,’ – should this be ‘a higher level of perceptual-cognitive skill’? Perception is really just the interpretations of sensory signals. I don’t think experts necessarily see anything differently, they just look in the right areas and interpret the information in a better way.

L34 – please check, it sounds odd.

L52 – CAVE is a form of virtual reality

L282 – I understand that definitions of smooth pursuit are difficult, but it would seem rigorous to justify your criteria with reference to previous work, rather than what would appear to be some arbitrarily chosen criteria.

Table 2 – should column 3 be called ‘invalid trials’ instead?

L440/441 – ‘the results of an intra-expert classification test to see whether inter-experts differences are smaller than than inter-class differences.’ I got confused here, although this may be my fault. Intra-expert classification test for inter-expert differences was confusing. Perhaps check this is the terminology you intended to use and also whether some of this terminology could be clearer. What is intra-expert? The within person variation for individuals in the expert group? (also typo with double than)

L447 – Again, is intra-expert the right word here? It sounds like you are comparing experts and intermediates, which would be inter-something, would it not?

L454 – intra instead of inter here, surely? Perhaps between-group/within-group terminology would be easier.

7. PLOS authors have the option to publish the peer review history of their article (what does this mean?). If published, this will include your full peer review and any attached files.

Reviewer #1: **Yes: **David Harris

---

## [Author Response · Author response to Decision Letter 1]

16 Apr 2021

We thank the reviewers for their constructive feedback, which helped to further improve the quality of our manuscript. In the following, we provide a point-by-point response.

Response to Reviewers:

1. “The use of references in the first few paragraphs of the introduction is quite infrequent, where one might expect background literature to be more regularly cited. Some more citations here about perceptual-cognitive expertise etc would be useful.”

A: We agree with the reviewer and added more related work to address perceptual cognitive expertise in the context of our work. (see line 3, 7, 8, and 78)

2. L13 – ‘a more highly developed perception,’ – should this be ‘a higher level of perceptual-cognitive skill’? Perception is really just the interpretations of sensory signals. I don’t think experts necessarily see anything differently, they just look in the right areas and interpret the information in a better way.

A: We corrected this statement in the manuscript accordingly (line 9). 

3. L34 – please check, it sounds odd.

A: We changed the wording of the sentence (line 24).

4. L52 – CAVE is a form of virtual reality

A: We adjusted the wording to “4k 360° video in HMDs,…” (line 36-37).

5. L282 – I understand that definitions of smooth pursuit are difficult, but it would seem rigorous to justify your criteria with reference to previous work, rather than what would appear to be some arbitrarily chosen criteria.

A:Since we only have the dispersion in pixels, it is even harder to find a reference, as there are no information about the size of a pixel of the HMD in mm. We therefore stick to the size of 100 px which corresponds to the typical length of the routes between the players.

6. Table 2 – should column 3 be called ‘invalid trials’ instead?

A:This point is also corrected in the revised version (see table 2, line 229-230).

7. L440/441 – ‘the results of an intra-expert classification test to see whether inter-experts differences are smaller than than inter-class differences.’ I got confused here, although this may be my fault. Intra-expert classification test for inter-expert differences was confusing. Perhaps check this is the terminology you intended to use and also whether some of this terminology could be clearer. What is intra-expert? The within person variation for individuals in the expert group? (also typo with double than)L447 – Again, is intra-expert the right word here? It sounds like you are comparing experts and intermediates, which would be inter-something, would it not?L454 – intra instead of inter here, surely? Perhaps between-group/within-group terminology would be easier.

A: We changed the wording to gaze variation between experts/ groups and hope to have made this point clearer now (line 339-354).

---

## [Editor Report · Decision Letter 2]

20 Apr 2021

Soccer Goalkeeper Expertise Identification based on Eye Movements

PONE-D-20-29324R2

Dear Dr. Hosp,

We’re pleased to inform you that your manuscript has been judged scientifically suitable for publication and will be formally accepted for publication once it meets all outstanding technical requirements.

Kind regards,

Greg Wood, PhD

Academic Editor

PLOS ONE
---

## [Editor Report · Acceptance letter]

23 Apr 2021

PONE-D-20-29324R2 

Soccer Goalkeeper Expertise Identification based on Eye Movements 

Dear Dr. Hosp:

I'm pleased to inform you that your manuscript has been deemed suitable for publication in PLOS ONE. Congratulations! Your manuscript is now with our production department. 

Kind regards, 

on behalf of

Dr. Greg Wood 

Academic Editor

PLOS ONE